# Cognitive-Emotional Benefits of Weekly Exposure to Nature: A Taiwanese Study on Young Adults

**Yin-Yan Yeung [1,\*] and Chia-Pin Yu [1,2]**

1    School of Forestry and Resource Conservation, National Taiwan University, Taipei 10617, Taiwan; simonyu@ntu.edu.tw
2    The Experimental Forest, National Taiwan University, Nantou 557009, Taiwan
\*    Correspondence: d06227101@ntu.edu.tw

**Abstract:** Empirical evidence of nature's benefits to cognitive and emotional well-being is emerging. In this study, 48 Taiwanese young adults (24 indoors and 24 outdoors in urban greenspace) completed four weekly 45 min exposure sessions. The study explores whether the outdoor group surpasses the indoor group in cognitive and emotional well-being and nature connectedness. There were no significant differences for the indoor group across different measurements of rumination and connectedness to nature. However, the outdoor group displayed a significant reduction in rumination post-test compared to the one week prior and the first session. Similarly, for sessions two, three, and four and one month post-test, the outdoor group's connectedness to nature was significantly higher than pre-test. Specific autobiographical memory was enhanced while overgeneral autobiographic memory was reduced during the third and fourth sessions, though these changes were not sustained at one-month follow-up. Surprisingly, both groups yielded similar results in decreased depression, anxiety, and stress. A significantly higher number of outdoor group participants had employed nature exposure for coping with stress or emotions after the program. We discuss the implications of this for counseling services for young adults and highlight future research possibilities, including formulating a nature-exposure protocol and a program evaluation for consolidating evidence-based nature prescription.

**Keywords:** nature; depression; ruminations; autobiographical memory; connectedness to nature

## 1. Introduction

Depression is the third leading cause of disease worldwide. Its onset ranges from mid-adolescence to middle age [1], and there is an increasing trend of depression internationally in the college student population [2]. In Taiwan, 15.1% of a community sample reported depressive symptoms, with younger age (those in an age group of 20 to 44) being a variable significantly related to a higher depression rate [3]. Apart from psychiatric and psychological interventions as treatments, systematic review and meta-analysis has revealed the effects of nature-based interventions on mental health [4], depression, and anxiety [5]. There is demand for more evidence-based support of the long-term outcomes and effects on behavior that nature-based programs produce [6], as well as demand for more evidence-based encouragement for people to be physically active in nature [7,8].

### 1.1. Identify Evidence-Based Outcome Measurements

Identifying outcome measurement accurately can result in better representation of what the benefits of nature exposure are. According to Bratman et al. [9], the production of affective benefits from nature experiences may occur through multiple psychological causal mechanisms and pathways, including decreased stress and negative affect, increased subjective well-being and positive affect via connectedness to nature, and regulation of affect as guided by cognitive processes.

The cognitive model of depression may help explain how depression develops and is maintained, and the model can provide insight regarding which outcome measurement is best for evaluating affect-regulation-related cognitive changes in nature, as was mentioned by Bratman et al. [9]. According to Beck [10–12], biased acquisition and processing of information play a primary role in the development and maintenance of depression. Latent schemas, which are internally stored representations of stimuli, ideas, and experiences, can be activated by internal events or external environmental events and can influence the processing of incoming information. Supported by neurobiological data, Disner et al. [13] highlighted how biased attention, memory, and thought are interrelated with biased information processing, which results from neurobiological malfunctioning. Attention to stimuli with a negative valence blocks out the processing of potentially more positive information. Ruminative thought patterns constantly remind the individual of their own perceived flaws. While biased memory is related to biased attention and processing, there is a difference in autobiographical memory recall (i.e., less specific but excessive generalization in memory) between depressed and non-depressed patients. Ample empirical evidence explains the relationships between depression and biased attention [14–16], biased memory [17–19], and ruminative thoughts [20,21], and through intervention, these biases can be modified [22–25].

In Beck's model, negative views held by depressed individuals about themselves, the world, and the future form a cognitive triad [10], and depression is instituted by one's view of oneself. Therefore, while Disner et al.'s model [13] can help identify cognitive constructs empirically related to depression, it is crucial to consider self-referential thinking, which entails an individual relating information from the external world to themselves. Therefore, in evaluating a nature-exposure program, it is worth identifying the cognitive variables, particularly those with self-referential properties, that nature may positively influence in people.

Robust findings of nature's impact on attention have mainly focused on restored or sustained attention [26–29], which do not specifically concern attention on oneself. Another impact nature may have on attention is attention shifting or attentional deployment [9]. Research indicates some aspects of natural environments allow for "positive distraction" away from the self [30–32] and attention driven specifically to the aesthetic qualities of nature [33–35]. Studies demonstrating a greater appreciation of the body's function or image after nature exposure [36,37] demonstrate how attention related to oneself can be shifted to be more positive because of the influence of nature. Evidence of whether exposure to nature can influence attention to the global self and others has yet to be established.

Rumination is focused attention to one's distress and carries a self-referential property. As for nature's impact on rumination, however, findings are still emerging. Nature exposure's impact on reducing rumination has not yet been confirmed as a cognitive domain in a meta-analysis [38] and currently is not viewed as an evidence-based mental health benefit [9]. Nevertheless, isolated findings indicate the effects. Among healthy participants, walking in nature reduced ruminative thoughts, while walking in an urban setting did not have the same result, whether the walk was 90 min [39] or as short as 30 min [40]. More data is needed to build an evidence base for the relationship between nature exposure and rumination.

Empirical evidence on memory centers around the relationship between working memory and exposure to nature, though the findings are not consistent [8,38,41–45]. Evidence, however, has not yet been established for a relationship between nature and autobiographical memory, a memory system that is a marker of depression and consists of episodes recollected from an individual's life. Given the presence of stress impairs the ability to access consolidated autobiographical memories [46], we expect overgeneral autobiographical memory would be lowered and specific autobiographical memory would be increased during exposure to nature. Nature's restorative effects [47,48] may render individuals less preoccupied by distressing thoughts; as a result, bottom-up information processing and, ultimately, specific retrieval of memory could be more likely.

Measurements of self-reflection, one of the cognitive domains, were not listed either in the available quantitative studies of nature or in Disner et al.'s model [13] of depression. The Attention Restoration Theory [48] addresses the ability of an environment to improve concentration and mental fatigue. Themes related to "reflective thinking" [49], "expanding personal perspective," and "inspiring a discovery of self" [50] have emerged from qualitative studies of nature experiences. A reduction in stress while in nature facilitates enhanced reappraisal capability [51] and mental flexibility [38]. Each of these potentially contributes to increased self-reflection and self-insight in nature. The resulting reappraisal counteracts the tunnel vision of depressed patients and, thus, may be beneficial to emotional functioning.

In short, work addressing treatment for depression is in demand for Taiwanese young adults. Both the latest views of nature's benefits and traditional views of depression consistently point to the importance of exploring its cognitive elements. The above literature review narrowed down the cognitive domains that may be improved during nature exposure, in addition to serving as an important guide for identifying evidence-based evaluation tools that may reflect cognitive changes in nature corresponding to affective problems, particularly those related to depression.

### 1.2. Define Evidence-Based Nature Experience

To measure nature's impact on human cognitive and emotional functioning, an evidence-based program is required. Bratman et al.'s [9] ecosystem service perspective provided a suitable reference for establishing program content.

Nature. Whereas biodiversity [9] and "being away" [47] are acknowledged as favorable aspects of natural settings for a healing experience, travelling distance may hinder individuals [52]. Though systematic review revealed urban greenspaces have mixed associations with a reduction in depression, in general, nature exposure has an association with positive emotions [53]. This favors hosting of a therapy program in a park to accommodate the busy lives of people living in urban areas.

Exposure. Exposure refers to the amount of contact with nature [9]. A systematic review of nature and human attention revealed the duration of exposure to environmental treatments has varied from 40 s to three hours, across one session or multiple sessions [38]. Two nature-exposure programs, one targeting ruminative thoughts [39] and one targeting psychological states [8], hosted walks in nature of 90 min and 50 min, respectively. Research focusing on the longitudinal effects and sustainability of nature's impact on adult health is in demand [6].

Experience. Cultivating appropriate sensory qualities is a priority for nature experiences, as is building interaction patterns [9]. Walking is the most frequently employed mode of physical engagement and exposure in nature [38]. Having participants complete the journey alone is a common approach in studying nature exposure and cognition and psychological state [8,39]. Hunter et al. [54] allowed participants to freely immerse in nature by walking, sitting, or doing both, adjusted time duration for each weekly experience, and suggested behaviors to avoid during the nature experience (e.g., physical exercise, phone calls, etc.). This approach addresses autonomy, a fundamental need linked to the health and psychological well-being of emerging adults [55,56], which should be regarded as an essential element in designing a nature experience.

Through identifying evidence-based outcome measurement and defining the program that has yielded desirable findings in prior research, we want to find the true differences in effect between greenspaces and the indoors on cognitive and emotional changes.

## 2. Methods

### 2.1. Study Hypothesis, Design and Content

We examined the cognitive and emotional effects for young adults of staying in urban greenspaces, with an assumption that greenspace is superior to an indoor environment in yielding positive effects. While our focus was on the difference between one week pre-

and one month post-test, we also took the measurement immediately after the weekly session in order to examine the trend of the changes as the sessions proceeded, which we anticipated should, for the outdoor group, show progressively enhanced positive indicators of well-being (such as attention to positive information or self-reflection and insight) and progressively lowering negative indicators (such as rumination or anxiety-depressive symptoms). Given nature experiences can change one's behavioral orientation towards nature, we wanted to explore whether there would be any difference between the two groups after the program in their number of visits to nature or their utilization of nature exposure for coping with emotions or stress.

Regarding nature selection, we chose Drunken Moon Lake, the largest and most easily accessible greenspace area at National Taiwan University (NTU). The path around the lake is nearly 480 m and has shaded grassland under the trees. The biodiversity of the park is rich, with ducks, geese, birds, fish, and squirrels easily seen despite the presence of manmade constructions such as stalls and teaching buildings. In general, this area grants a sense of serenity and connection to nature. As for the indoor group serving as a comparison group, due to the pandemic and entry restrictions in most faculties, we chose a forestry building that allowed for quality monitoring. The venue is a four-story building with corridors shorter than 100 m and a conference room in which the participants could freely move around during the program sessions. Lights and air ventilation were activated during sessions. To make the environment less congested for the indoor group, we restricted the number of participants to eight per session. Each session took place on a weekday and was conducted at 11:30 a.m. for the outdoor group and 1:00 p.m. for indoor group to avoid the peak hours of the usage of the park and the building, correspondingly. See Figures 1 and 2 for images from the indoor and outdoor groups, respectively.

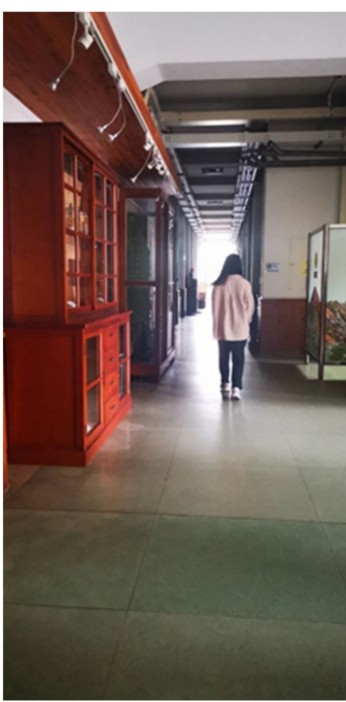

**Figure 1.** Picture of Indoor Group.

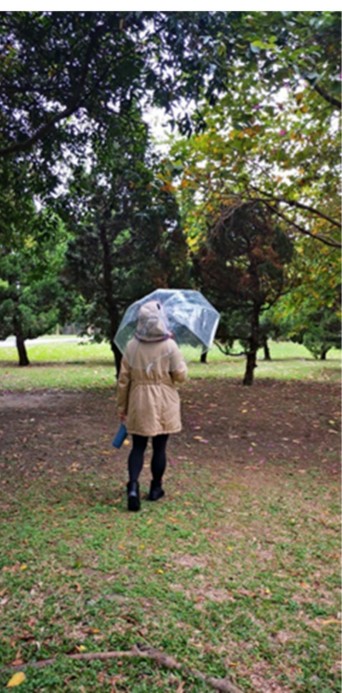

**Figure 2.** Picture of Outdoor Group on Rainy Days.

For the exposure, we arranged for four weekly 45 min visits to observe the impact of exposure to the same greenspace over a series of sessions. Some degree of flexibility was allowed for the participants to schedule the weekly sessions, that is, at most two days earlier or later than the original schedule, to accommodate their routine. As a result, the sessions were spaced out over an interval of five to nine days. For the experience, walking, sitting alone, or doing both were allowed, while it was recommended not to sleep, conversate, use a cell phone, perform physical exercise, or eat. To reduce interference by insects, outdoor group participants were advised to wear pants and long sleeves. A raincoat, transparent umbrella, and mosquito repellent were provided so participants could appreciate the environment in any conditions. Two sessions were rescheduled, one due to heavy rain and one due to chilly weather.

The participants were gathered at the entrance of the largest park on the NTU main campus for the outdoor group and in front of the forestry building for the indoor group. All participants had to silence their phones and set a 45 min alarm before starting a session. Both groups completed questionnaires onsite immediately after the session. The indoor group completed the questionnaire in the conference room, and the outdoor group completed it in the park (or a covered area near the park on rainy days) with a writing pad provided. The participants could earn $1300 Taiwanese dollars upon completion of the four participation sessions of the program, the one-week-pre-test and one-month-post-test sessions, and an online questionnaire one week after the fourth session. The right to withdraw from the study was established, while confidentiality of all participant information was strictly upheld. Briefing was conducted, with a consent form signed before administration of the pre-test (see Figure 3 for the study's flowchart).

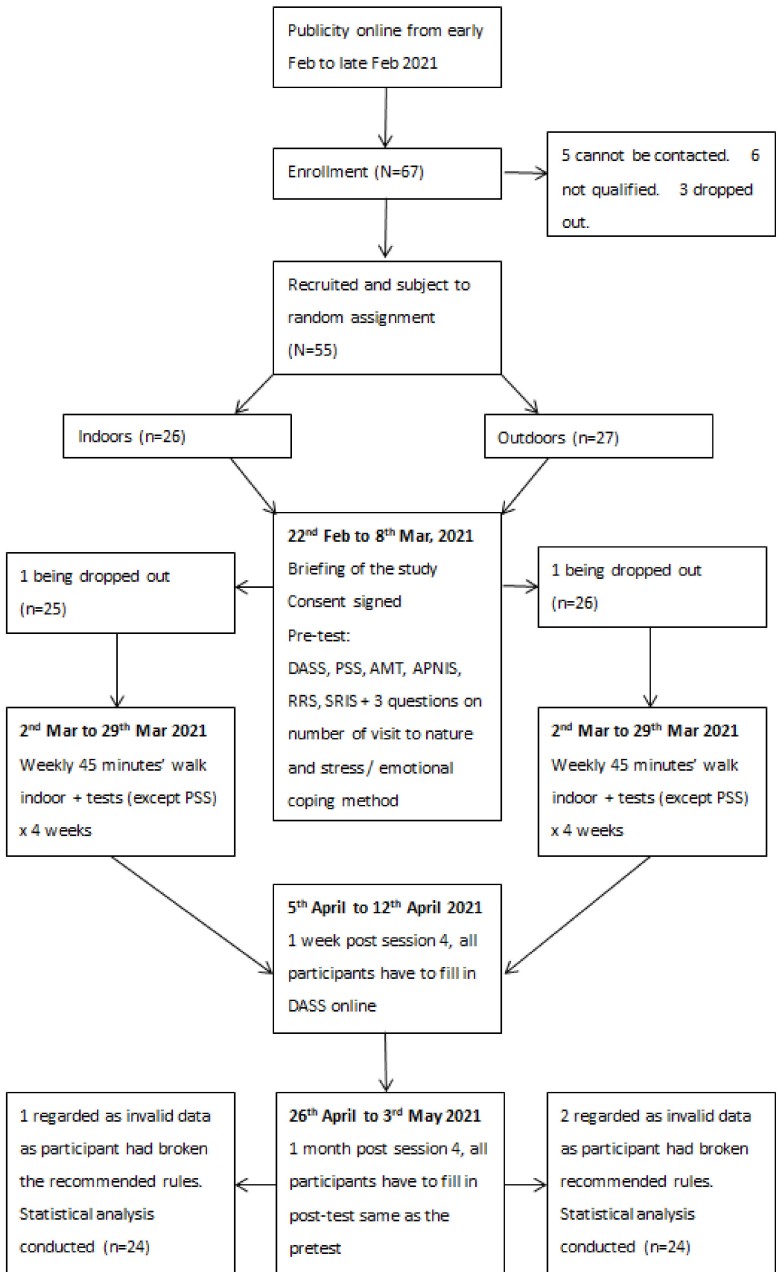

**Figure 3.** Flowchart of The Study.

We used an online messaging platform to remind the participants a day before each session or test. One member of the research team was present in the park or the forestry building for observation of any abnormalities or non-compliance with the recommendations of the program. They observed one participant depart from the park and two participants use their phones. These three cases were not included in the final evaluation. The study was approved by Ethical Review of National Taiwan University (Case no.: 202008HS011).

*2.2. Participants*

A statistical power analysis was performed using GPower 3.1.9.7. [57] to estimate sample size for a repeated-measures ANOVA. A medium (Cohen's d = 0.50) effect size assuming alpha = 0.05 and power = 0.80 yielded a total sample size of 22. University students of age above 21 were recruited from an online social media platform. In total, 48 participants managed to complete all measurements and the four sessions of the program without breaching the behavioral recommendations. The number of participants is regarded

as sufficient to yield the intended statistical effect. Regarding demographics, 52.1% of the participants were male (mean age = 25.08 years, SD = 2.20), and 47.9% were female (mean age = 25.91 years, SD = 5.66). The majority of participants came from NTU (81.3%), while the others represented six other universities. Most were undergraduates, with 64.6% studying in Bachelor's programs, while 35.4% were in Master's programs. Random assignment to indoor and outdoor groups, with insignificant differences in age [t(46) = 1.35, $p > 0.05$], sex [$\chi^2(1) = 0.08$, $p > 0.05$], and affiliated academic programs (i.e., Bachelor vs. Master) [$\chi^2(1) = 0.82$, $p > 0.05$] are noted.

*2.3. Measurements*

The Depression Anxiety Stress Scale—21 items (DASS-21) (1 week pre-test, immediately after each session, 1 week and 1 month post-test), self-rating last week's situation on a four-point Likert scale, was used to measure the emotional states of depression, anxiety, and stress [58]. DASS-21's total score reflects general or affective distress. The three-factor structure of the scale has been validated for nonclinical [59] and clinical studies [60]. Its suitability for regular assessment and treatment evaluation has been established [61]. For the depression, anxiety, and stress subscales, Cronbach's alphas of 0.84, 0.79, and 0.83 were yielded, respectively. Good factor structure (CFA = 0.959, RMSEA = 0.063) for the Taiwanese college student sample [62] and a convergent validity with the Chinese Beck Depression Inventory and the Chinese State-Trait Anxiety Inventory in a Chinese college student sample [63] are reported. In this study, we employed the adapted Chinese scale [64]. Because the questionnaire was to evaluate the conditions of the previous week, we arranged an online evaluation one week after session four to reflect the impact of session four.

The Perceived Stress Scale—10 items (PSS-10) (1 week pre-test, 1 month post-test), originally a 14-item scale (PSS-14), measures the perception of global levels of stress in the past month and is applicable for any population [65,66]. Its shorter version, PSS-10, is comprised of 10 items rated on a five-point Likert scale and is reported to be of a superior psychometric property [67,68]. Its high validity has been reported in assessing perceived stress among Chinese adolescents or adults, with Cronbach's alpha varying from 0.79 to 0.83 and a concurrent validity on indices of anxiety and depression [69,70]. The present study employed the Chinese version of PSS by Chu and Kao [71].

The rumination subscale of the Ruminative Response Scale (RRS) (1 week pre-test, immediately after each session, and 1 month post-test), a subscale of the Response Styles Questionnaire [72], consists of 22 items that assess repetitive, self-focused thought processes about the meanings, causes, and consequences of one's negative affect (e.g., "Think why I always react this way" and "Think about how alone you feel"). The items are rated on a four-point Likert scale ranging from 1 (almost never) to 4 (almost always), with higher sums of scores indicating greater rumination. The RRS has high internal consistency (0.97) [73] and test–retest reliability (r = 0.67) [74]. It had been employed in a study involving cognitive measures of the impact of nature viewing [75]. Studies have demonstrated the ability of the scale to capture within-person variation in depressive brooding [76] and real changes in rumination over a two-month period [77]. The present study employed the translated and validated 13-item symptom-based rumination subscale, with high concurrent validity with Beck Depression Inventory (r = 0.79, $p < 0.001$) in the Taiwanese population [78].

The Autobiographical Memory Test (AMT) (1 week pre-test, immediately after each session, and 1 month post-test) [79] was employed to assess specific or overgeneral autobiographical memory, which are considered cognitive vulnerability factors and concurrent depressive symptoms [80,81]. The AMT has been administered to adolescents [82,83] and young adults [84,85]. It differentiated in a Taiwanese population the depressed from the non-depressed, who reported fewer specific and more categorical autobiographical memories [86].

In AMT, respondents are presented with positive and emotional cue words and are asked to recall and describe personal memories of which those cue words remind them [79]. With consent obtained, we adapted the online version of the Autobiographical Memory Test

(AMT) [79]. We followed Hallford et al.'s approach [87] (p. 898), in that "the participants were not explicitly advised to recall specific AMs, but rather just to recall a personal event from their life that could not be from the past week or have been mentioned multiple times. They were asked to state as many details as they could in relation to the event but were not given a definition of a specific AM or practice cues." In addition, there was no time limit for how long participants had to respond to each cue word. This approach for investigating AMT seems more valid and more sensitive to individual differences in non-clinical samples that tend to have a greater proportion of specific memories [88,89].

Addressing the cultural uniqueness of cue words, we generated a bank of 60 total positive and negative words by referencing Asian studies [19,86,90] and Western studies [87,91,92]. Repeated words were deleted. The final emotional or adjective word bank for positive words was 25 words, and for negative words, it was 28 words. Six additional positive words (achieved, beautiful, committed, free, funny, and grateful) and two additional negative words (fear and vigilance) were added by searching online discussion platforms and Chinese studies of emotions [93]. Each set of the AMT test was 10 words, 5 positive and 5 negative, in alternate order. The six sets of tests had previously been trial-tested on 10 young adults to ascertain the comprehensibility of the instructions and words. Coding was performed based on established criteria [79]. Twenty responses (200 items in total) were coded as specific, categorical, extended, or semantic by two coders who were at least of graduate qualification in psychology. The intraclass correlation coefficient (ICC) of 0.88 for the two independent scorers should be regarded as reflection of excellent agreement among the two coders [94]. As such, one of the coders completed all the residual ratings. We treated AMT as a unifactorial structure [90] and added the positive and negative items together for analysis. We conceptualized overgeneral AM as including categorical and extended AM [95], so we summed these two items to obtain the overgeneral AM.

The scale of Attention to Positive and Negative Information (APNI) (1 week pre-test, immediately after each session, and 1 month post-test) [96] is 40 items. It is rated on five-point Likert scale for cognitive tendencies (including attend to, think about, and focus on) for either positive (Attention to Positive Information, API; 22 items) or negative information (Attention to Negative Information, ANI; 18 items). Samples of the items are "I pay attention to things that lift me up" and "I notice when something is not going well even if it's a trivial thing." Attention to positive information (Cronbach's alpha = 0.84) was positively correlated with positive affectivity and negatively correlated with negative affectivity, whereas attention to negative information (Cronbach's alpha = 0.72) was positively correlated with negative affectivity [96]. The Cronbach's alphas of the API subscale and APN subscale were 0.87 and 0.84, respectively, in a Chinese sample [97]. The present study employed its translated version [97].

The Self-Reflection and Insight Scale (SRIS) (1 week pre-test, immediately after each session, and 1 month post-test) [98], which is 20 items in total, is comprised of self-reflection (SRIS-SR) and insight (SRIS-IN) subscales. Its sample questions are: "It is important to me to try to understand what my feelings mean" (self-reflection) and "Thinking about my thoughts makes me more confused" (insight). The test–retest reliability of its seven-week period was 0.77 (SRIS-SR) and 0.78 (SRIS-IN). The SRIS-SR positively correlated with anxiety and stress, while the SRIS-IN negatively correlated with depression, anxiety, and stress. The present study employed the 12-item SRIS-Chinese (SRIS-C) scale that was found to have Cronbach's alphas of 0.79, 0.87, and 0.83 for total, SRIS-SR (seven items), and SRIS-IN (five items), respectively, as well as a three-week test–retest reliability of 0.74 among Taiwanese college students [99].

The Connectedness to Nature Scale (CNS) (1 week pre-test, immediately after each session, and 1 month post-test) assesses the "experiential sense of oneness with the natural world" of individuals [100] (p. 504) or the cognitive identity dimension of one's relationship with nature [101]. Being a 14-item inventory rated on a five-point Likert scale, CNS is of a single factor and possesses high internal consistency ($\alpha = 0.84$) and test–retest reliability ($r = 0.79$) [100]. Its inverse correlation with perceived stress ($r = -0.16$, $p = 0.01$), anxiety

(r = −0.11, $p$ = 0.04), and depression (r = −0.15, $p$ = 0.04) [102] as well as its positive correlation with subjective well-being [34] have been reported. Because CNS can be utilized for evaluating whether interventions can increase a person's contact with nature [100], it is a suitable indicator of whether our program design could successfully manipulate nature exposure. The present study employed Li and Cao's [103] translated and validated CNS, which had a Cronbach's alpha of 0.90.

Frequency of visits to nature and emotional or stress coping method (1 week pre-test and 1 month post-test) are constructed items in this study for investigating changes in participants' daily interactions with nature, which could be an indicator of motivation to continuously seek connection with nature and should be favorable for well-being. One week before and one month after the study, we required participants to specify their number of 45 min visits to nature by themselves or with family and friends in the past month and to provide three emotion or stress coping methods they had employed. After studying the descriptive statistic of frequency of visits, we recoded the frequency as 0 to 4, corresponding with those numbers of visits, and 5 if the frequency of visits was five or more. Meanwhile, we studied the emotional coping methods listed by the participants and treated the answer as 0 or 1, dependent upon whether nature exposure as an emotional coping method was present in the past month.

We acknowledge there was concern for some of the assessments, such the scale of APNI and SRIS, being administered immediately after the session, given it is uncertain whether self-referential thinking could be transformed immediately after a session. The same concern applied for DASS, as the instrument assesses the condition of the past week, and measurement immediately after the session may not reflect the effects potentially caused by the session. Nevertheless, to make the assessment easier to complete during the pandemic when physical gathering of the participants was not encouraged, we arranged all questionnaires to be administered at once after each session. The difference between one week pre-test and one month post-test was our main focus, while each session's data served as a reference to monitor the tendency of changes. It was an explorative effort, given that APNI and SRIS had not been employed in empirical studies of nature experiences as far as we were aware. For the same reason of simplifying the study protocol during the pandemic, we limited our evaluations to one week pre-test, four program sessions, and one month post-test (without taking one week post-test), as we wanted to explore the impacts of nature exposure at a longer duration. A one-time brief online post-test evaluation aimed only at evaluating the impact of the fourth session of nature exposure on DASS-21 was administered that asked participants to report their situation over the past last week.

### 2.4. Method of Analysis

The data were analyzed in SPSS statistics 27. We calculated the reliability, mean, and standard deviation of all the measurements. Two-way repeated-measures ANOVA was performed for all measurements, except PSS, because it only had pre-test and post-test measurements, in order to examine the main effect as well as the interaction the effect of time and group. For the repeated measures, we adjusted the degrees of freedom to Greenhouse–Geisser (when Epsilon of Greenhouse–Geisser was <0.75) or Huynh–Feldt (when Epsilon of Greenhouse–Geisser was >0.75) when the test of sphericity was significant [104]. For AMT, we produced scale data arising from frequency counts. For DASS total scale and subscales, the scales were square-root transformed to meet the assumptions of normality and constant variance. A visual inspection of model residuals was used to test whether model assumptions were being met. A less stringent LSD test was employed for the DASS depression subscale, for which no significant pairwise comparison was yielded in the multivariate test by using Bonferroni. Across all other pairwise comparisons, we employed Bonferroni in the subscales.

For PSS, which was measured only at one week pre-test and one month post-session, ANCOVA was performed. The effectiveness of the indoor vs. outdoor group in reducing PSS was compared with the difference in the pre-PSS score treated as a covariate. Finally,

we conducted *t*-test to compare the indoor and outdoor groups' frequency of paying 45 min visit to nature in the last month as well as their employment of nature for stress or emotional coping.

## 3. Results

### 3.1. Reliability, Means, and Standard Deviations of the Measurements

Table 1 lists the reliability, means, and standard deviations of the measurements broken down into indoor and outdoor groups. As noted, the anxiety subscale of DASS and the negative information subscale of APNI had Cochbach's Alphas lower than 0.80 for a number of the measurements. The reliability of all other scales was within or approached the good-to-excellent range.

**Table 1.** Reliability, means, and standard deviations of the measurements (N = 48) by indoor and outdoor.

| | Time | Reliability | Indoor | | Outdoor | |
|---|---|---|---|---|---|---|
| | | | M | SD | M | SD |
| DASS_depression (7 items) | 1 week pre | 0.82 | 4.50 | 4.63 | 4.92 | 4.042 |
| | Session 1 | 0.86 | 3.67 | 4.08 | 4.17 | 3.71 |
| | Session 2 | 0.92 | 3.00 | 4.33 | 3.88 | 5.10 |
| | Session 3 | 0.87 | 4.38 | 4.77 | 3.46 | 3.40 |
| | Session 4 | 0.90 | 3.63 | 4.81 | 2.92 | 3.30 |
| | 1 week post | 0.89 | 3.04 | 4.28 | 3.38 | 4.23 |
| | 1 month post | 0.90 | 4.25 | 5.19 | 3.08 | 3.61 |
| DASS_anxiety (7 items) | 1 week pre | 0.78 | 5.00 | 4.28 | 3.67 | 3.13 |
| | Session 1 | 0.72 | 4.71 | 3.78 | 3.17 | 2.79 |
| | Session 2 | 0.68 | 4.46 | 3.48 | 2.38 | 2.10 |
| | Session 3 | 0.77 | 3.17 | 2.57 | 3.25 | 3.66 |
| | Session 4 | 0.77 | 3.13 | 3.48 | 2.54 | 2.96 |
| | 1 week post | 0.82 | 2.96 | 3.13 | 2.92 | 4.16 |
| | 1 month post | 0.80 | 3.67 | 3.50 | 3.46 | 3.79 |
| DASS_stress (7 items) | 1 week pre | 0.78 | 7.71 | 4.2 | 6.79 | 4.48 |
| | Session 1 | 0.84 | 6.88 | 4.46 | 6.25 | 4.6 |
| | Session 2 | 0.81 | 8.08 | 4.88 | 5.42 | 3.35 |
| | Session 3 | 0.88 | 6.29 | 4.18 | 5.71 | 5.40 |
| | Session 4 | 0.90 | 6.21 | 5.31 | 4.33 | 4.10 |
| | 1 week post | 0.86 | 5.54 | 4.46 | 4.92 | 4.13 |
| | 1 month post | 0.86 | 5.92 | 4.34 | 6.21 | 4.77 |
| DASS_Total (21 items) | 1 week pre | 0.91 | 17.21 | 11.86 | 15.38 | 10.38 |
| | Session 1 | 0.91 | 15.25 | 11.00 | 13.58 | 9.55 |
| | Session 2 | 0.90 | 15.54 | 11.17 | 11.67 | 8.16 |
| | Session 3 | 0.93 | 13.83 | 10.18 | 12.42 | 11.22 |
| | Session 4 | 0.95 | 12.96 | 12.95 | 9.79 | 9.63 |
| | 1 week post | 0.94 | 11.54 | 10.93 | 11.21 | 11.82 |
| | 1 month post | 0.93 | 13.83 | 11.27 | 12.751 | 10.77 |
| PSS (14 items) | 1 week pre | 0.87 | 30.21 | 7.97 | 30.63 | 6.23 |
| | 1 month post | 0.89 | 28.58 | 7.13 | 30.21 | 6.97 |
| RRS_rumination (13 items) | 1 week pre | 0.85 | 27.17 | 7.43 | 28.54 | 7.46 |
| | Session 1 | 0.89 | 25.38 | 8.69 | 27.88 | 6.04 |
| | Session 2 | 0.89 | 27.04 | 9.02 | 24.58 | 6.09 |
| | Session 3 | 0.94 | 27.42 | 10.24 | 24.71 | 8.97 |
| | Session 4 | 0.88 | 25.75 | 10.40 | 23.29 | 6.50 |
| | 1 month post | 0.91 | 25.29 | 8.53 | 23.45 | 7.79 |

**Table 1.** *Cont.*

| | Time | Reliability | Indoor | | Outdoor | |
|---|---|---|---|---|---|---|
| | | | **M** | **SD** | **M** | **SD** |
| APNI_positive information (22 items) | 1 week pre | 0.78 | 56.38 | 6.85 | 56.33 | 7.13 |
| | Session 1 | 0.87 | 55.58 | 9.21 | 57.29 | 6.77 |
| | Session 2 | 0.89 | 54.00 | 8.56 | 56.13 | 9.52 |
| | Session 3 | 0.89 | 55.75 | 8.70 | 54.67 | 9.89 |
| | Session 4 | 0.89 | 55.42 | 7.98 | 57.29 | 8.51 |
| | 1 month post | 0.88 | 56.54 | 7.70 | 56.17 | 9.42 |
| APNI_negative information (18 items) | 1 week pre | 0.66 | 37.96 | 5.30 | 39.17 | 6.06 |
| | Session 1 | 0.69 | 37.96 | 6.00 | 39.75 | 5.39 |
| | Session 2 | 0.73 | 38.46 | 5.05 | 38.04 | 6.64 |
| | Session 3 | 0.75 | 38.33 | 5.96 | 38.25 | 6.74 |
| | Session 4 | 0.73 | 37.75 | 5.70 | 37.46 | 6.56 |
| | 1 month post | 0.77 | 37.58 | 6.61 | 38.83 | 6.55 |
| SRIS_self-reflection (7 items) | 1 week pre | 0.89 | 29.71 | 7.421 | 32.58 | 5.80 |
| | Session 1 | 0.89 | 29.96 | 6.85 | 33.00 | 5.93 |
| | Session 2 | 0.88 | 30.51 | 6.89 | 31.67 | 5.87 |
| | Session 3 | 0.90 | 28.55 | 6.61 | 32.29 | 6.43 |
| | Session 4 | 0.96 | 28.13 | 8.03 | 32.50 | 6.84 |
| | 1 month post | 0.88 | 28.42 | 7.45 | 32.29 | 5.53 |
| SRIS_insight (5 items) | 1 week pre | 0.75 | 20.54 | 4.81 | 21.71 | 5.06 |
| | Session 1 | 0.80 | 20.25 | 5.12 | 21.25 | 4.37 |
| | Session 2 | 0.87 | 19.50 | 5.73 | 22.46 | 4.99 |
| | Session 3 | 0.78 | 20.92 | 4.54 | 21.00 | 4.56 |
| | Session 4 | 0.87 | 20.38 | 5.39 | 23.04 | 4.53 |
| | 1 month post | 0.90 | 22.46 | 4.70 | 22.71 | 5.84 |
| CNS (14 items) | 1 week pre | 0.83 | 48.96 | 8.46 | 50.29 | 9.18 |
| | Session 1 | 0.87 | 50.42 | 8.00 | 53.17 | 9.07 |
| | Session 2 | 0.85 | 49.38 | 7.65 | 53.79 | 8.47 |
| | Session 3 | 0.89 | 50.79 | 7.85 | 54.04 | 10.39 |
| | Session 4 | 0.89 | 50.08 | 8.10 | 55.63 | 9.35 |
| | 1 month post | 0.93 | 49.92 | 8.37 | 56.79 | 11.39 |
| AMT_specific | 1 week pre | N.A. | 3.71 | 1.92 | 4.42 | 2.06 |
| | Session 1 | N.A. | 2.50 | 1.79 | 2.96 | 2.37 |
| | Session 2 | N.A. | 2.92 | 1.86 | 3.63 | 1.71 |
| | Session 3 | N.A. | 2.79 | 1.69 | 4.33 | 1.86 |
| | Session 4 | N.A. | 2.46 | 1.61 | 4.96 | 2.27 |
| | 1 month post | N.A. | 3.75 | 2.33 | 4.08 | 1.79 |
| AMT_over-general | 1 week pre | N.A. | 5.83 | 2.06 | 4.92 | 1.82 |
| | Session 1 | N.A. | 6.50 | 2.02 | 5.92 | 2.32 |
| | Session 2 | N.A. | 5.33 | 2.51 | 4.58 | 2.04 |
| | Session 3 | N.A. | 6.13 | 1.83 | 4.38 | 1.61 |
| | Session 4 | N.A. | 6.63 | 1.84 | 4.25 | 1.89 |
| | 1 month post | N.A. | 4.79 | 1.77 | 4.79 | 1.59 |
| Frequency of visit to nature with family/friends for 45 min (1 single item) | 1 week pre | N.A. | 2.33 | 1.86 | 1.25 | 1.33 |
| | 1 month post | N.A. | 1.33 | 1.24 | 1.71 | 1.46 |
| Frequency of visit to nature on one's own for 45 min (1 single item) | 1 week pre | N.A. | 1.63 | 1.35 | 1.63 | 1.50 |
| | 1 month post | N.A. | 1.79 | 1.69 | 2.58 | 2.17 |
| Emotional or stress coping by nature exposure (1 single item) | 1 week pre | N.A. | 3 | N.A. | 7 | N.A. |
| | 1 month post | N.A. | 6 | N.A. | 13 | N.A. |

Abbreviations: M, Mean; SD, Standard Deviation; N.A., Not Applicable; DASS, Depression Anxiety Stress Scale; PSS, Perceived Stress Scale; RRS, Ruminative Response Scale; APNI, The scale of Attention to Positive and Negative Information; SRIS, The Self-Reflection and Insight Scale; CNS, Connectedness to Nature Scale; AMT, Autobiographical Memory Test.

### 3.2. Repeated Measurement of Cognitive or Emotional Well-Being

Table 2 lists the effects of the time and time x group on the instruments that underwent repeated measurement analysis.

**Table 2.** Repeated measurement of cognitive or emotional well-being (N = 48).

| Measurement | Effect of Time | | | Effect of Time x Group | | |
|---|---|---|---|---|---|---|
| | **F** | **df** | $\eta^2$ | **F** | **df** | $\eta^2$ |
| RRS_rumination | 3.19 * | 4.40, 202.35 | 0.07 | 2.38 * | 4.40, 202.35 | 0.05 |
| AMT_specific | 4.82 ** | 5, 230 | 0.10 | 3.54 ** | 5, 230 | 0.07 |
| AMT_over-general | 3.87 ** | 4.88, 224.44 | 0.08 | 2.91 * | 4.88, 224.44 | 0.06 |
| APNI_positive information | 1.34 | 5, 230 | 0.03 | 1.51 | 5, 230 | 0.03 |
| APNI_negative information | 1.04 | 4.52, 208.04 | 0.02 | 0.25 | 4.52, 208.04 | 0.03 |
| SRIS_self-reflection | 1.02 | 5, 230 | 0.02 | 1.31 | 5, 230 | 0.03 |
| SRIS_insight | 3.12 * | 3.83, 176.17 | 0.06 | 3.83 | 3.83, 176.17 | 0.05 |
| DASS_depression | 2.90 * | 5.73, 263.44 | 0.06 | 0.93 | 5.73, 263.44 | 0.02 |
| DASS_anxiety | 4.97 ** | 5.67, 260.91 | 0.10 | 1.27 | 5.67, 260.91 | 0.03 |
| DASS_stress | 4.58 ** | 5.75, 264.70 | 0.09 | 1.46 | 5.75, 264.70 | 0.03 |
| DASS_total | 7.10 ** | 4.35, 200.22 | 0.33 | 0.60 | 4.35, 200.22 | 0.07 |
| CNS | 4.83 * | 3.56, 163.81 | 0.10 | 2.83 * | 3.56, 163.81 | 0.06 |

Abbreviations: RRS, Ruminative Response Scale; AMT, Autobiographical Memory Test; APNI, The scale of Attention to Positive and Negative Information; SRIS, The Self-Reflection and Insight Scale; DASS, Depression Anxiety Stress Scale; CNS, Connectedness to Nature Scale. * indicates $p < 0.05$; ** indicate $p < 0.01$.

#### 3.2.1. Rumination Subscale of RRS

For the rumination subscale of RRS, there was a significant effect of time, as well as an interaction effect of time and group. For the indoor group, the pairwise comparison did not indicate a significant reduction in rumination across the different time points, whereas for the outdoor group, a significant reduction in rumination was observed when comparing the post-test (M = 23.45, SE = 1.67) to one week pre-test (M = 28.54, SE = 1.52, t(47) = −3.38, $p < 0.01$) and the first session (M = 27.88, SE = 1.53, t(47) = −2.89, $p < 0.01$). See Figure 4 for comparison of the two groups across the different time points.

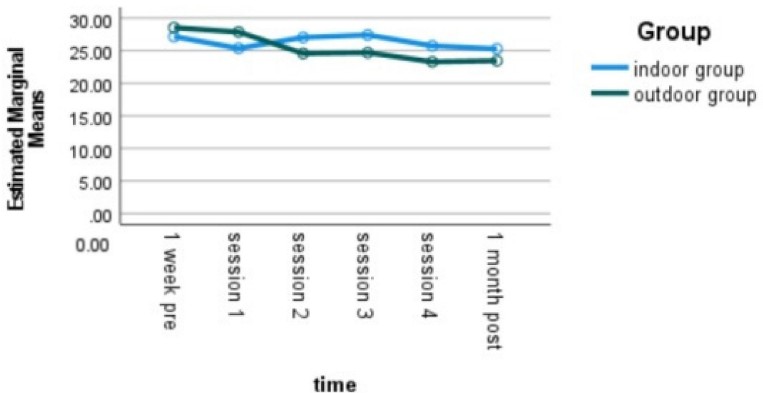

**Figure 4.** Estimated marginal means of rumination subscale of Ruminative Response Scale by groups across different times.

#### 3.2.2. Specific AMT and Overgeneral AMT

Repeated measures revealed there was a significant main effect of time and interaction effect of time and group on the number of specific AM. In the pairwise comparison of groups, the number of specific AM of the outdoor group was significantly higher than that of the indoor group for session three (indoor group mean = 2.79, SE = 0.36, and outdoor group mean = 4.33, SE = 0.36, t(47) = 3.01, $p < 0.01$) and session four (indoor group mean = 2.46, SE = 0.40, outdoor group mean = 4.96, SE = 0.40, t(47) = 4.39 $p < 0.01$). When examining

the two groups individually, the indoor group's number of specific AM had no significant differences across any time point. However, for the outdoor group, at session four, the number of AMT specific memories (M = 4.96, SE = 0.46) was significantly higher than for session one (M = 2.96, SE = 0.48, t(23) = 3.58, $p < 0.01$) and for session two (M = 3.63, SE = 0.37, t(23) = 3.11, $p < 0.01$).

A significant time and time x group interaction effect is reported for the number of overgeneral AM. For both the indoor and outdoor groups, individual group profiles indicate there were not any significant pre- or post-test differences in the number of overgeneral AM. In addition, the outdoor group had no significant differences in the number of overgeneral AM pre-test when compared with the indoor group (indoor group mean = 5.83, SE = 0.42, outdoor group mean = 4.92, SE = 0.37, t(47) = 1.64, $p > 0.05$). However, the outdoor group had a significantly lower mean number of overgeneral AM than the indoor group did in session three (indoor group mean = 6.13, SE = 0.37, outdoor group mean = 4.38, SE = 0.33, t(47) = −3.52, $p < 0.01$) and in session four (indoor group mean = 6.63, SD = 0.38, outdoor group mean = 4.25, SD = 0.39, t(47) = −4.41, $p < 0.01$).

### 3.2.3. APNI Subscales and Self-Reflection Subscale of SRIS

For the APNI subscales and self-reflection subscale of SRIS, there were not any significant time or time $\times$ group interaction effects. However, for the insight subscale of SRIS, there was a significant time effect. The pairwise comparison indicates the insight scale post-test score (M = 22.58, SE = 0.77) was significantly higher than the scores for session one (M = 20.75, SE = 0.69, t(47) = 3.69, $p < 0.01$) and for session three (M = 20.96, SE = 0.66, t(47) = 4.01, $p < 0.01$).

### 3.2.4. DASS

For the DASS total and all its subscales, there was no interaction of time x group, but there was a significant time effect. For the DASS depression subscale, session two (M = 3.44, SE = 0.68, t(47) = −2.13, $p < 0.05$), session four (M = 3.27, SE = 0.59, t(47) = −2.32, $p < 0.05$), one week after session four (M = 3.21, SE = 0.61, t(47) = −2.56, $p < 0.05$), and one month after session four (M = 3.67, SE = 0.66, t(47) = −2.14, $p < 0.05$) had significantly lower means than the pre-test (M = 4.71, SE = 0.62). Interestingly, repeated measures also showed session four (M = 2.83, SE = 0.46, t(47) = −3.32, $p < 0.01$) and one week post-session-four (M = 2.94, SE = 0.53, t(47) = −3.04, $p < 0.01$) were significantly lower than the pre-test for the anxiety subscale (M = 4.33, SE = 0.55). Similar findings were yielded for the stress subscale, with session four (M = 5.27, SE = 0.69, t(47) = −3.02, $p < 0.01$) and one week post-session-four (M = 5.23, SE = 0.61, t(47) = −3.43, $p < 0.01$) being significantly lower than the pre-test score (M = 7.25, SE = 0.62). This was also the case for the total scale, with session four (M = 11.38, SE = 1.65, t(47) = −2.81, $p < 0.01$) and one week post-session-four (M = 11.38, SE = 1.63, t(47) = −2.72, $p < 0.01$) being lower than the pre-test score (M = 14.42, SE = 1.48).

### 3.2.5. CNS

For CNS, there was a significant main effect of time as well as interaction effect of time and group. In the pairwise comparison of groups, the CNS of the outdoor group was significantly higher than the CNS of the indoor group for session four (indoor group mean = 50.08, SE = 1.79, outdoor group mean = 55.63, SE = 1.91, t(23) = 2.19, $p < 0.05$) and one month post-session-four (indoor group mean = 49.92, SE = 1.71, outdoor group mean = 56.79, SE = 2.32, t(23) = −2.38, $p < 0.05$). Examining the simple effect of time, the indoor group had no significant difference across different times; however, the outdoor group's CNS scores for session two (M = 53.79, SE = 1.73, t(23) = 3.32, $p < 01$), session three (M = 54.04, SE = 2.12, t(23) = 2.68, $p < 0.05$), session four (M = 55.63, SE = 1.91, t(23) = 3.45, $p < 0.01$), and one month post (M = 56.79, SE = 2.32, t(23) = 3.78, $p < 0.01$) were each significantly higher than the pre-test score (M = 50.29, SE = 1.88). See Figure 5 for comparisons of the two groups across the different time points.

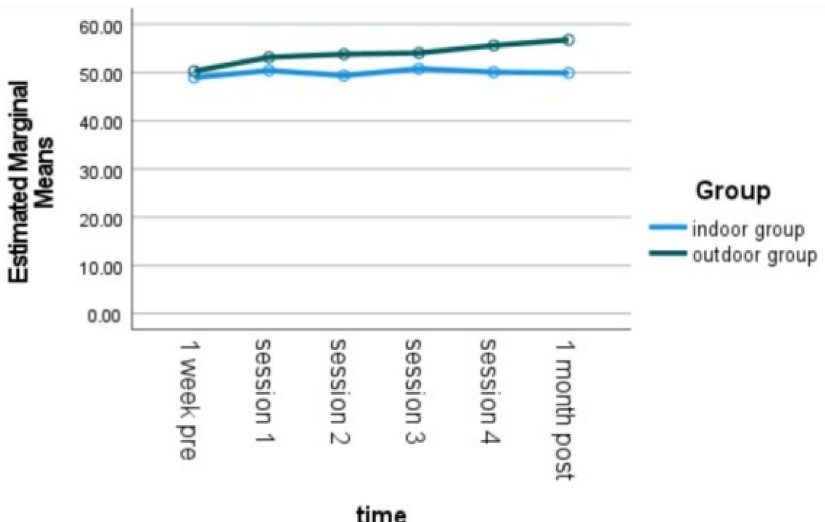

**Figure 5.** Estimated marginal means of the Scale of Connectedness to Nature by groups across different times.

*3.3. ANCOVA on PSS*

For PSS, the assumption of linearity was fulfilled (R = 0.65 between pre- and post-test PSS score), while the homogeneity of the regression slope was confirmed by the Univariate test of GLM (Group*PSS_pre F (1, 44) = 0.86, $p > 0.05$). The result of ANCOVA suggests the two groups had no difference in lowering PSS (F (1, 45) = 0.78, $p > 0.05$). Table 3 illustrates the findings of the pairwise comparison.

**Table 3.** Mean Differences in post-test PSS score by Group Controlling for pre-test PSS score (N = 48).

| Comparison | Mean Difference (Outdoor–Indoor) | $p$ | Standard Error | Bonferroni Adjusted 95% CI |
|---|---|---|---|---|
| Indoor vs. outdoor group | 1.35 | >0.05 | 1.53 | $-1.73 \rightarrow 4.43$ |

Comparisons based upon ANCOVA adjusted means controlling for pre-test PSS score mean of 30.42. Abbreviation: PSS, Perceived Stress Scale.

*3.4. T-Test on Frequency of Visits to Nature and Chi-Square on Emotional & Stress Coping Method*

Regarding the frequency in the past month of 45 min visits to nature with family or friends, the indoor and outdoor groups differed significantly one week pre-test (indoor group mean = 2.33, SE = 0.38, outdoor group mean = 1.25, SE = 0.27, t(47) = $-2.33$, $p < 0.05$) but insignificantly post-test (indoor group mean = 1.33, SE = 0.25, outdoor group mean = 1.71, SE = 0.30, t(47) = $-0.96$, $p > 0.05$). For the frequency of 45 min visits to nature by oneself, the indoor and outdoor groups did not differ significantly either at one week pre-test (indoor group mean = 1.63, SE = 0.28, outdoor group mean = 1.63, SE = 0.31, t(47) = 0.00, $p > 0.05$) or at one month post-test (indoor mean = 1.79, SE = 0.35, outdoor mean = 2.58, SE = 0.44, t(47) = $-1.41$, $p > 0.05$). For the presence of employing nature for stress or emotional coping, the two groups differed insignificantly pre-test (indoor group mean = 3, outdoor group mean = 7, $\chi^2$ (1) = 2.02, $p > 0.05$) but differed significantly post-test (indoor group mean = 6, outdoor group mean = 13, $\chi^2$ (1) = 4.27, $p < 0.05$).

## 4. Discussion

The present study investigated several confirmed or potential cognitive and emotional benefits humans gain from exposure to nature. In our findings, among the cognitive variables, ruminations and AMT were two domains that differentiated the indoor group from the outdoor group. We found the outdoor group, who spent 45 min weekly in nature for four weeks, had lowered ruminations one month post-test, while the indoor group did not have the same change. This result echoes findings reported by Bratman et al. [39]

and Lopes et al. [40] regarding ruminative thoughts being lowered via exposure to nature. Meanwhile, in our study, the pre- and post-test differences in the counts of specific and overgeneral AM were not significant for the indoor and outdoor groups. The outdoor group's specific AM was significantly higher and its overgeneral AM was significantly lower than the indoor group's for sessions three and four. Theoretically biased thoughts, including memories and ruminations, influence processing of incoming information and play a primary role in the development and maintenance of depression [13], and depression is empirically associated with rumination [20,21] and specific/overgeneral AM [17–19]. Therefore, exposure to nature for four 45 min sessions successfully reduces rumination and results in a desirable impact on AM, as found in the present study, and should be regarded as favorable for decreasing depression.

In the present study, we attempted to explore using nature and non-nature exposure to address self-referential-thinking-related responses associated with attention. We employed the scale of Attention to Positive and Negative Information as an indicator. We found indoor and outdoor groups showed no significant difference for pre-test, post-test, and all sessions. One possible explanation of why this program that lasted for only four sessions did not produce significant effects in this domain is that cognitive tendencies, such as optimism and pessimism, that emphasize the positive or negative aspects of life events are associated with less changeable trait factors [96]. Our current findings do not rule out how, in other studies, attention has been reactive in positive ways to nature. Seemingly, the best method to assess and measure whether attention is shifted or deployed by nature exposure is the following: first, discern whether nature can direct our attention to positive environmental stimuli at all, and second, explore whether repeated exposure to nature helps humans to more easily transfer their attention to a more positive self-referential stimulus.

Surprisingly, there was no evidence in the present study of nature exposure offering great capacity in enhancing self-reflection and insight. Reflective thinking and self-discovery have been common themes in qualitative studies of the subjective experience of exposure to nature [49,50]. The failure of this study to yield similar positive findings may indicate that, compared with the seemingly more inspiring forest or wilderness utilized in the two referenced qualitative studies, the urban green environment in which this study was conducted did not possess stimuli meaningful enough to stimulate reflection or insight.

In this study, we employed DASS (which covers clinical symptoms of depression, anxiety, and stress) and PSS (which measures subjective experiences of stress) to robustly confirm the effects on emotional well-being. The findings consistently indicate the two groups did not differ one month post-test in mood or stress-related measures. DASS provided additional information that both groups exhibited similar effects, decreasing depression, anxiety, and stress at the fourth session and one week after the fourth session. The decrease in depression was sustained one month after the last session.

The above findings on affective and emotional well-being, including mood and stress, are quite different than most of our literature review, which showed contact outdoors with nature elements is better at enhancing mood than staying indoors (see review [105]). We offer the following explanations for our findings. First, because of the pandemic, our indoor group had to be conducted in the forestry school's building for convenience. Although it is a standard classroom setting, it does have some forestry decoration that may elicit a certain emotional healing effect. Second, although the 45 min sessions of walking and sitting did not explicitly require practicing mindfulness skills, participants may have attained some degree of mindfulness simply because they refrained from using phones and instead focused attention on a new environment they may have viewed as fascinating. In addition, the participants, though not compulsorily required, were advised not to sleep during the program. The indoor group, staying in a relatively monotonous environment, may have taken more initiative to walk to avoid feeling sleepy, which would have resulted in them engaging in more physical exercise. Given that mindfulness [24,106] and physical exercise [107,108] are known to decrease depression and to protect against the emergence of depression, the indoor group's possible increase in physical exercise and mindfulness may

have resulted in mood benefits that offset the boost in emotional well-being from nature exposure that was experienced by the outdoor group.

Of importance, the outdoor group experienced a greater sense of connectedness to nature at the fourth session and one month post-test than the indoor group. This finding is comparable to a prior finding that in a student population, nature experiences can lead to an enhanced nature connection for up to six weeks [109]. Given that CNS has previously associated favorably with a well-being index [34,102], the present finding is another indicator of the benefits of nature exposure. Empirical data on the long-term impact on connectedness to nature resulting from interventions are rare, and our study helps contribute to the established evidence.

We explored changes in participants' interaction patterns with nature by asking about their frequency of visiting nature alone or with family and friends, as well as their usage of nature exposure for coping with emotions or stress. We note the result for the outdoor group in the increase in the number of visits to nature with family and friends should be interpreted with caution. First, under random assignment, the outdoor group had a significantly lower pre-test frequency, which implied a different baseline, which was not favorable for this statistical analysis. Second, while the outdoor group did not more frequently visit nature alone, the increase in the number of its visits to nature with family and friends from pre-test to post-test was instead beyond our expectation, because the pandemic should have hindered motivation to visit nature with other people. We wonder if our standard for the time of the visit, which was set as 45 min, was too demanding, particularly during the pandemic; perhaps we should have attempted to capture nature visits of a shorter duration. We could have instead referenced Hunger et al.'s [54] findings on the impact of 20–30 min nature experiences on stress reduction.

While the program only lasted four weeks for the young adults, the lowered rumination and increased connectedness to nature were sustained one month after the program. In addition, there are data showing the outdoor group's significantly higher utilization of nature exposure for coping with emotions or stress. A series of visits to urban greenspace during the experiment seemingly mobilized participants to visit nature when they felt negative emotions and, thus, may have resulted in a continuous increase in the participants' connections with nature and a decrease in their rumination. Such evidence of nature's longer-term cognitive benefits and inducing continuous nature connection brings good news to counseling services for young adults, given that the nature-visiting approach is somewhat more like a leisure activity, which can make young adults less resistant to try it when compared with traditional counseling services. In addition, we show that the healing place might not necessarily be wilderness but can be a park nearby, while the visit can be as brief as only four 45 min sessions. This, from both the recipient and policy maker's point of view, is appealing given that such brief intervention in urban greening is convenient and of lower transportation or other setup cost. Connecting individuals with nature for lowering of rumination is a sound approach, as the world at present is constantly facing large-scale health hazards and subsequent social distancing, which can trigger repetitive negative thinking and hinder traditional emotional support from family and friends. This is particularly true for young adults who are facing stress at the turning points of their life for career development and family establishment.

The strengths of the study are as follows. First, this was a random-assignment study that comprehensively employed measurements of cognitive domains containing self-referential elements, of which the rumination subscales of RRS and AMT were found to be capable of differentiating the indoor group versus the outdoor group. Second, the program lasted four sessions, and we attempted to observe the cumulative effects of nature exposure over a longer term, specifically one month after the program. Third, we referred to Bratman et al.'s [9] ecosystem service perspective to develop a nature-exposure protocol defining nature, exposure, and experience. Considering young adults prioritize their autonomy, we structured the program with flexibility, allowing participants to freely choose a route within the park and requiring no designated behavior other than a few basic

restrictions. The weekly visits to nature, which occurred for four weeks, were personally adjusted (scheduled earlier or postponed within a two-day range) to enhance the motivation of participants to attend all sessions. We provided a raincoat, transparent umbrella, and mosquito spray to make the trip comfortable for the participants and to facilitate their connection with nature. This protocol possibly increased enjoyment of the trip, cognitive benefits during and after the sessions, and motivation after the program to use nature experiences for stress or emotional coping.

There are a number of limitations of the study. First, because of the pandemic, we were unable to schedule the indoor group anywhere other than in our own department, the forestry school. The department, with a relatively woodsier display than other departments, may have facilitated a degree of nature connectedness that contaminated the differences between the indoor and the outdoor group. Second, because of practical concerns during the pandemic, we sacrificed a one-week-post-test result for a longer-term measurement, the one-month-post-test result. Third, originally, we aimed to identify self-inferential thinking using the cognitive theory; however, unfortunately, inventories that serve this purpose are limited. The scales for attention and self-reflection/insight employed in this study may not be responsive to variations in self-referential cognition. Fourth, in order to avoid making either exposure experience too congested, we scheduled the two groups to two different time slots according to the degree of usage of the venues. Though the timeslots were both around lunch hour, we still have to admit that this could be a variable affecting the treatment effect. Fifth, due to the pandemic, the sample in the experiment was not large enough to include a treatment-free control group this time (that is, indoor or outdoor without non-suggested behavior). Lastly, the total number of subjects completing all sessions being 48 hindered further exploration of the association between coping with emotions or stress and a continual decrease in rumination and a rising connectedness to nature, as was interestingly found in the present study.

Conducting a similar study on clinically depressed patients, using a larger sample size, exploring cognitive-emotional impacts of nature exposure over a longer time period, further examination of the ideal duration for an impactful nature exposure, studying the relationship of behavioral changes (e.g., motivation to visit nature or employ nature exposure to cope with emotions or stress) and well-being after repeated nature exposure, and enriching protocol of nature experiences by developing dos and don'ts for a program are possible future research topics. Our findings of similar effects in the indoor group and the outdoor group in decreasing depression, anxiety, and stress indicate it is worth investigating the ways manmade buildings with nature-related decorations facilitate therapeutic effects. This could be timely and beneficial, as in the future, pandemics may again keep people indoors and present mental health challenges.

## 5. Conclusions

The cognitive theory posits that environmental stimuli cause changes in emotions and information processing. We studied whether four 45 min sessions of nature exposure could result in favorable changes in cognition and mood. The present study, though, did not find differences in mood between the indoor group and the outdoor group; however, the results did indicate nature exposure was more effective for reducing ruminations, positively impacting autobiographical memory, and connecting young adults to nature. Although we discussed why the indoor group experienced the benefit of mood improvement, we cannot rule out that, if the program duration was longer, the favorable impacts of relatively smaller cognitive changes could be transformed to impact overall emotional well-being, given that, logically, the impact of an environment on depressed mood takes time. Ultimately, through defining nature, exposure, and experience, we managed to establish an evidence-based yet flexible, convenient, comfortable, and appealing urban greenspace-exposure program for young adults. This can serve as a reference for nature prescription, a value-for-money treatment modality, and a therapeutic focus worldwide in the coming decade.

**Author Contributions:** Conceptualization, Y.-Y.Y.; Formal analysis, Y.-Y.Y.; Funding acquisition, C.-P.Y.; Methodology, Y.-Y.Y.; Project administration, Y.-Y.Y.; Supervision, C.-P.Y.; Writing—original draft, Y.-Y.Y.; Writing—review & editing, Y.-Y.Y. and C.-P.Y. All authors have read and agreed to the published version of the manuscript.

**Funding:** This research was funded by Ministry of Science and Technology of Taiwan, grant number MOST 110-2410-H-002-146-MY3.

**Institutional Review Board Statement:** The study was approved by Ethical Review of National Taiwan University (Case no.: 202008HS011).

**Informed Consent Statement:** Written consent had been obtained from all participants.

**Data Availability Statement:** Original data is available at https://doi.org/10.6084/m9.figshare.19351331.

**Acknowledgments:** We thank the participants for their participation in the study while they experienced active stress during the pandemic. We are grateful for Chun-Yen Chang, Yu-Sen Chang, Li-Ju Chen and Ye-Jen Lin's valuable advice on the conceptualization and program design. We acknowledge the kind assistance by Huan-Tsun Chen and Jittakon Ramanpong in data collection, Jordan Green in English editing, Chun-Lin Chan in online questionnaire design, Gary Lou in inter-rater reliability testing, Nicole Chu in participants' recruitment, and Tsz-kit Ko for reference list editing. Finally, we also want to express our gratitude to James Colee for his valuable opinions on the statistical analysis.

**Conflicts of Interest:** The authors declare that the research was conducted in the absence of any commercial or financial relationships that could be construed as a potential conflict of interest.

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
