# Peer review of "Cognitive-Emotional Benefits of Weekly Exposure to Nature: A Taiwanese Study on Young Adults"

_sustainability, doi:10.3390/su14137828_

Round 1
Reviewer 1 Report
The article provides very interesting and novel information in the given area of research on managing emotional disorders by means of nature exposure. Research covers both new self-referential elements involved in the well-known cognitive model of depression and the circumstances caused by the COVID-19 pandemic. The research looks conceptually well-grounded taking into consideration up-to-date knowledge on depression. Self-reflection, as one of important cognitive domain involved in emerging of depressive mode, is not well elaborated in the aspect of managing depressive conditions in sub-clinical persons with nature therapy. So, the empirical data obtained seem to be useful and actual. The research done fills the gap between the practice of psychological aid and evidence-based interventions by nature exposure, differentiating the important circumstances for improved efficacy and long-term outcomes. The important evidence, as I suppose, is that nature experience significantly better impacts the definite cognitive variables – ruminations and autobiographical memory, and due to this investigation, we get the idea which sings of emotional disorders could be better recuperated with nature exposure. The data give the path to further investigations, as it has been questioned the different impact of wild nature and urban green zones with big biodiversity on other elements of depressive triad - reflective thinking and self-discovery. I would recommend publishing the paper as it is. The idea is clearly written, all elements are presented well, the investigation, results, discussion, and conclusions match the requirements of proper scientific product.
Author Response
Dear Reviewer,
Thank you very much for your positive comments to our manuscript. Wanna to share with you that we are indeed very excited when the statistical result comes out. Though it does not totally fit our hypothesis (as most of the research finding does), we are happy to find with an evidence-based design we can generate a program that can bring effect of longer term on cognitive-emotional well-being. Thank you for giving us a credit that our research can differentiate the important circumstances for improved efficacy and long-term outcomes. And as you have captured one of our discussion, we also treasure very much the present finding can stimulate us to think about the role of biodiversity at wild land on self-reflection. Hope we, from different corner of the world, can work bit by bit to find the natural pills for those suffered from depression.
We are now editing the manuscript according to the suggestions by other reviewers. These includes adding conclusive remarks for the lengthy introduction, adding a section concerning statistical method, and also some other minor formatting issue. We hope we can successfully publish this paper very soon.
Once again, thank you for your sharing, support and encouragement!
Best regards,
Y.Y.Yeung & Simon Yu
Reviewer 2 Report
The paper is very interesting, the subject matter is original, and the experimental nature is a strength. The use of proven research tools and providing a study flowchart is also an advantage. The following comments concern the structure of the paper and editing issues.
1 The abstract should be improved. The number of participants is not given, nor is the division into outdoor and indoor groups described. The abstract in this form gives the impression of a discussion piece without a precise description of the methodological assumptions and specific results of the study.
2 I would also suggest to indicate already in the abstract or title that this is a Taiwanese study.
3 The introductory section is very extensive. No clearly stated purpose derived from this introduction.
4 A description of the procedure of the current study could in turn already be part of the section on methods
Citation convention inconsistent with journal requirements, similar to APA style. MDPI requires references numbered in order of appearance in the text, not alphabetic order and author names inside.
Please, check the instruction for a list of recommended parts and numbering rules. There are too many parts, it should rather be a hierarchical structure. The beginning part is not numbered at all. Moreover, it’s also recommended to name section 3 as “results” not “findings”.
7 I don't see any part describing methods of analysis, which should be in the material and methods section. There is only a reference to software at the beginning of the description of results.
8 Both figures have a very narrow scale range on the vertical axis. This is a generally criticized manipulation of the results to highlight the differences. Please try changing the scale range starting from zero.
Finally as minor editing issue, I feel that double spaces between words are common in the text, which is difficult to check without the word version.
Author Response
Manuscript ID: sustainability-1759810
Authors: Yin-yan Yeung * , Chia-Pin Yu
Response to Reviewer 2 Comments
Dear Reviewer,
Thank you very much for your valuable comments. Since the corrective marks are quite a lot, I suggest you to adopt all changes in the word file and then read those marked in yellow.
Point 1: The abstract should be improved. The number of participants is not given, nor is the division into outdoor and indoor groups described. The abstract in this form gives the impression of a discussion piece without a precise description of the methodological assumptions and specific results of the study.
Response 1: The abstract had been edited according to your advice (see page 1 marked in yellow). We originally plan to add some statistical figures. However due to the word limits (not more than 200 words, now it is of 206 words) the further expansion of the abstract is hindered. See if you think the present presentation is fine. Your further suggestions are very welcome.
Point 2: I would also suggest to indicate already in the abstract or title that this is a Taiwanese study.
Response 2: Done (see page 1 marked in yellow). Title now changed to: Cognitive-Emotional Benefits of Weekly Exposure to Nature: Its A Cognitive-Emotional Benefits on Taiwanese Study on Young Adults. Abstract also mentioned the young adults are Taiwanese.
Point 3: The introductory section is very extensive. No clearly stated purpose derived from this introduction.
Response 3: A conclusive remark has been added for the introduction (see page 3 marked in yellow) while subheadings had been added for the two major focus of the introduction (namely 1.1. Identify Evidence-based Outcome Measurements, 1.2. Define Evidence-based Nature Experience) (see page 1 & 3).
Point 4 & 7: A description of the procedure of the current study could in turn already be part of the section on methods. I don't see any part describing methods of analysis, which should be in the material and methods section. There is only a reference to software at the beginning of the description of results.
Response 4 & 7: A section “Methods” added (see page 9: 2.4 Method of Analysis). Procedure of the experiment now reframed as 2.1. Study Hypothesis, Design and Content (see page 3 marked in yellow).
Point 5: Citation convention inconsistent with journal requirements, similar to APA style. MDPI requires references numbered in order of appearance in the text, not alphabetic order and author names inside.
Response 5: Citation conversion conducted on the original APA reference style of the manuscript (see page 18 – 24 marked in yellow).
Point 6: Please, check the instruction for a list of recommended parts and numbering rules. There are too many parts, it should rather be a hierarchical structure. The beginning part is not numbered at all. Moreover, it’s also recommended to name section 3 as “results” not “findings”.
Response 6: Numbering rules followed. All parts got numbered, with hierarchical numbering adopted.
Point 8: Both figures have a very narrow scale range on the vertical axis. This is a generally criticized manipulation of the results to highlight the differences. Please try changing the scale range starting from zero.
Response 8: The scale range has been changed to a zero start (see figures at page 13 & 14)
Point 9: Finally as minor editing issue, I feel that double spaces between words are common in the text, which is difficult to check without the word version.
Response: Double spaces deleted.
Rating:
Are the research design, questions, hypotheses and methods clearly stated?
( ) ( ) (x must be improved) ( )
Response: Study hypothesis is elaborated (see page 3 to 4 under 2.1. Study Hypothesis, Design and Content). Subtitle added for Methods (namely 2.1. Study Hypothesis, Design and Content, 2.2. Participants, 2.3. Measurements, 2.4. Method of Analysis) (see page 3 - 9). A section of “Methods of Analysis” added (see page 9 – 10).
For empirical research, are the results clearly presented?
( ) (x can be improved) ( ) ( )
Response: By adding a section of “Method of Analysis” the result part of the manuscript is shortened. We have added subtitle for each measurement’s statistical analysis (see page 12 – 15). This, together with our minor editing of the result we hope the readers can digest the result much more easily.
Is the article adequately referenced?
( ) (x can be improved) ( ) ( )
Response: Since the reference list is now already of 109 items at this stage we do not further add reference to the list. Please advice if you think at what part of the manuscript should be further referenced. Thank you very much!
Other remarks:
- Sorry to tell that we find we have in some of t-test’s report SD instead of SE (i.e. standard error) We now have corrected them as SE. We have already countered checked the numerical data.
- For the first paragraph of discussion (page 15) we have deleted “our review of AMT revealed previous reporting of the superiority of exposure to nature in increasing specific memory and decreasing overgeneral memory; however” because the content is an incorrect one.
We hope the above responses had answered your concerns. Should there be any further advice please feel free to tell us. Thank you very much!
Best regards,
Y.Y.
Reviewer 3 Report
Dear Author(s),
The manuscript entitled “Weekly Exposure to Nature: Its Cognitive-Emotional Benefits on Young Adults” deals with an interesting and current topic. I have only a few minor concerns beside one main issue that need to be addressed.
There are only two content issues that need to be explained/handled. First, why were the two groups treated at different times of the day? This could cause differences in the effect of treatments (this should be explained and at least admitted among the limitations). Second and most importantly, to gather reliable data, a control group should have been used that received no treatment at all.
Minor issues: There are a lot of unnecessary spaces between sentences all along the text. In line 81, the term “meta-analysis” should be used instead of “metal-analysis”. The formatting of citations and the references list is not in line with the journal’s requirements. The quality (resolution) of Figure 3 should be increased. The names of tests in lines 290 and 330 should be highlighted similarly to the other tests. A space is missing in line 300. An empty line is needed under Table 2 and 3. In line 439 only the one month post value of indoor group is mentioned, but that of outdoor group also should be mentioned for comparison. A comma in line 488 is unnecessary, it should be in the next line.
Author Response
Manuscript ID: sustainability-1759810
Authors: Yin-yan Yeung * , Chia-Pin Yu
Response to Reviewer 3 Comments
Dear Reviewer,
Thank you very much for your valuable comments. Since the corrective marks are quite a lot, I suggest you to adopt all changes in the word file and then read those marked in yellow.
Major issues
Point 1. Why were the two groups treated at different times of the day? This could cause differences in the effect of treatments (this should be explained and at least admitted among the limitations).
Response 1: We aim at finding a time convenient to the young adults who are busily engaged by classes while also taking into account desirability of the environment (i.e. noise and congestion). We chose lunch hour to increase the enrollment. For the Park it is the first half of the lunch time when students are still taking meals. For the forestry building it is the second half of the lunch time as most of the students should have left the building after settling necessary business after classes. We have admitted it as a limitation of the study at page 21 (marked in yellow).
Point 2: Second and most importantly, to gather reliable data, a control group should have been used that received no treatment at all.
Response 1: From our perspective, human is either inside or outside a building and therefore we can regard the indoor group as a control to the outdoor group. But similar to your idea we have considered to design a another control groups which is outdoor but not in nature (e.g. at road of urban) or staying indoor or outdoor without following the recommended behavior (i.e. not talking, not eating, no use of mobile phone). Unfortunately the pandemics had affected the enrollment and we do not have a large sample to further break down the sample to include these control groups. We have now admitted such limitation at page 21’s discussion.
Minor issues:
Point 3: There are a lot of unnecessary spaces between sentences all along the text.
Response 3: Deleted the spaces.
Point 4: In line 81, the term “meta-analysis” should be used instead of “metal-analysis”.
Response 4: Corrected.
Point 5: The formatting of citations and the references list is not in line with the journal’s requirements.
Response 5: Done
Point 6: The quality (resolution) of Figure 3 should be increased.
Response 6: Done
Point 7: The names of tests in lines 290 and 330 should be highlighted similarly to the other tests.
Response 7: Done
Point 8: A space is missing in line 300. An empty line is needed under Table 2 and 3.
Response 8: Done
Point 9: In line 439 only the one month post value of indoor group is mentioned, but that of outdoor group also should be mentioned for comparison.
Response 9: Done
Point 10: A comma in line 488 is unnecessary, it should be in the next line.
Response 10: Done
Rating:
Are the research design, questions, hypotheses and methods clearly stated?
( ) ( ) (x must be improved) ( )
Response: Study hypothesis is elaborated (see page 4 under 2.1. Study Hypothesis, Design and Content, marked in yellow). Subtitle added for Methods (namely 2.1. Study Hypothesis, Design and Content, 2.2. Participants, 2.3. Measurements, 2.4. Method of Analysis) (see page 3 - 9). A section of “Methods of Analysis” added (see page 9 – 10).
For empirical research, are the results clearly presented?
( ) (x can be improved) ( ) ( )
Response: By adding a section of “Method of Analysis” the result part of the manuscript is shortened. We have added subtitle for each measurement’s statistical analysis (see page 12 – 15). This, together with our minor editing of the result we hope the readers can digest the result much more easily.
Other issues:
- Sorry to tell that we find we have in some of t-test’s report SD instead of SE (i.e. standard error) We now have corrected them as SE. We have already countered checked the numerical data.
- For the first paragraph of discussion (page 15) we have deleted “our review of AMT revealed previous reporting of the superiority of exposure to nature in increasing specific memory and decreasing overgeneral memory; however” because the content is an incorrect one.
We hope the above responses had answered your concerns. Should there be any further advice please feel free to tell us. Thank you very much!
Best regards,
Y.Y.Yeung & Simon Yu